# NT5C1B Improves Fertility of Boar Spermatozoa by Enhancing Quality and Cryotolerance During Cryopreservation

**DOI:** 10.3390/ani15243530

**Published:** 2025-12-08

**Authors:** Shibin Wang, Lei Shi, Zhaoyang Zhang, Junjie Liu, Jiandong Xing, Jingxian Yang, Jiaxin Duan, Bugao Li, Guoqing Cao

**Affiliations:** 1College of Animal Science, Shanxi Agricultural University, Jinzhong 030801, China; wang0907bin@163.com (S.W.); sxausl@gmail.com (L.S.); moluwubian@163.com (Z.Z.); liujunjie6600@126.com (J.L.); 17678048462@163.com (J.X.); 15129168320@163.com (J.Y.); djxsxnydx@sxau.edu.cn (J.D.); 2Shanxi Key Laboratory of Animal Genetics Resource Utilization and Breeding, Jinzhong 030801, China

**Keywords:** NT5C1B, boar sperm, fertility, cryopreservation, cryo-tolerance

## Abstract

Based on a systematic evaluation of post-thaw characteristics, this study categorizes boar sperm into two distinct cryo-survival phenotypes: cryo-tolerant and cryo-sensitive. The freeze-tolerant group exhibited significantly superior post-thaw quality, including better motility, plasma membrane integrity, acrosome integrity, DNA integrity, and mitochondrial activity. Furthermore, these sperm demonstrated stronger antioxidant capacity and reduced oxidative damage (lower MDA levels). Through proteomic analysis, we identified seven key proteins associated with cold tolerance (e.g., HSPA12A, PRDX4). Functional validation revealed that supplementing the freezing extender with 1 µg/mL recombinant NT5C1B protein significantly improved multiple quality parameters in freeze-sensitive sperm, confirming its role as an effective factor in enhancing cryo-survival. In contrast, ADA protein supplementation showed no significant improvement. The well-established freezing protocol developed in this study provides additional options for boar semen cryopreservation.

## 1. Introduction

Semen cryopreservation represents the optimal strategy for achieving long-term storage of mammalian sperm [1,2]. Cryopreservation technology serves as a key driver, fundamentally transforming the practice of artificial insemination in the livestock industry. Currently, frozen bovine semen has been widely applied for fixed-time artificial insemination (AI) [3]. However, cryopreserved boar semen has limited application in artificial insemination within the swine industry, with liquid storage at 16–18 °C remaining the conventional approach [4]. The impairment of motility and fertility in boar spermatozoa resulting from cryopreservation procedures constitutes the primary factor limiting their commercial application [5,6,7]. Since the first successful boar sperm cryopreservation and AI with frozen/thawed semen in the 1950s, researchers have made great efforts to minimize cryoinjury to boar spermatozoa [8]. Based on this objective, the research team systematically applied and optimized various cryopreservation protocols and freezing media to improve the post-thaw motility and overall quality of boar sperm.

The strategic implementation of cryoprotectants in conjunction with optimized freeze-thaw protocols has been demonstrated to significantly improve post-thaw sperm quality parameters in most mammalian species. However, researchers found that the motility of post-thawed sperm is various in the individuals with similar fresh semen quality and genetic characteristics [9,10]. It is indicated that the quality of cryopreserved spermatozoa also largely relies on their strong tolerance to low temperature and resilience to withstand freezing-thawing treatments [11]. Because boar semen needs to be centrifuged before freezing, it is generally believed that there are two reasons for the differences in the post-thawed sperm motility of boar semen with the similar ejaculate quality. First of all, the difference of sperm proteins related to boar genetics leads to different sperm freezing tolerance [12]. Secondly, it was the freezing-thawing procedures that changed some structural and functional sperm proteins and resulted in the difference of their low temperature tolerance [13,14]. Based on this consideration, special attentions have been paid to the sperm proteins related to cryotolerance and their alterations during cryopreservation. Proteomic analyses have characterized the dynamic changes in boar sperm proteins throughout the freeze-thaw process, revealing specific proteins implicated in cryopreservation outcomes. It was confirmed the abundance of some proteins (TPI, SOD1, AWN, Clusterin, ODF2, Spermadhesins, ACE, AKAP3 and AKAP4) was significantly different between fresh and post-thawed boar semen [15,16,17,18]. These proteins were associated with key biological processes including sperm integrity, oxidative phosphorylation, and mitochondrial function [19]. Although a number of differential proteins during boar semen cryopreservation were identified, the mechanism of sperm proteomic modifications changes affecting their motility and function is still unknown. With the development of omics technology, more and more potential biomarkers related to sperm freezability in pig sperm have been investigated [20,21]. Currently, although studies have confirmed that osteopontin improves the fertilization rate and quality of frozen-thawed bovine epididymal sperm [22], no research has demonstrated that the sperm protein enhances the cold tolerance of boar sperm during semen cryopreservation.

Therefore, we employed proteomic technology to screen and identify the protein NT5C1B associated with freeze tolerance from boar sperm, verified the impact of this key differential protein on the freeze tolerance of boar sperm, and evaluated its effects on sperm fertilization capacity and embryonic development potential through in vitro fertilization and embryo culture [23]. This study plays a crucial role in germplasm resource utilization and enhancing animal fertility.

## 2. Materials and Methods

All animal experimental procedures in this study were conducted in accordance with the Guidelines for the Care and Use of Laboratory Animals issued by the Ministry of Science and Technology of the People’s Republic of China (Beijing, China, 2006), and were approved by the Institutional Animal Care and Use Committee of Shanxi Agricultural University [7].

### 2.1. Animals and Semen Collection

This study utilized semen samples collected from 37 healthy *Large White boars* (18–36 months old) provided by Shanxi Swinebaba Breeding Pig Co. Ltd., (Taiyuan, China). Immediately after collection, the semen samples were diluted with Beltsville Thawing Solution [24] at a 1:1 (*v*/*v*) ratio post-collection. The diluted semen was then examined microscopically to assess sperm concentration, progressive motility, and morphological abnormalities. Semen samples meeting the criteria of sperm motility > 80% and concentration > 2 × 10^8^/mL were considered qualified. Following freeze-thaw procedures, the 9 boars with the highest post-thaw sperm motility were classified as the cold-tolerant group, while the 9 boars with the lowest post-thaw sperm motility were designated as the cold-susceptible group.

### 2.2. Semen Cryopreservation and Thawing

Semen freezing is performed according to a modified freezing scheme [25]. Initially, following initial dilution, the semen sample was centrifuged (800× *g*, 10 min). After discarding the supernatant, the collected sperm pellet was then reconstituted in lactose-egg yolk (LEY) extender containing 11% lactose solution and egg yolk in a 4:1 (*v*/*v*) ratio. The sperm suspension was subjected to a controlled cooling protocol, gradually reducing the temperature to 4 °C over 2 h. The cooled sample was then combined with a freezing extender (1:1, *v*/*v*) consisting of LEY supplemented with 2% Orvus Es Paste and 6% glycerol, and cryotolerance-associated proteins (NT5C1B and ADA). Load the samples into an automatic filling machine for precise filling and sealing. Semen samples were cryopreserved using a programmable freezer (Model CJ-L37, Haier Biomedical, Shenzhen, China) following a specific stepwise protocol. The freezing regimen consisted of an initial cooling phase from 4 °C to −5 °C at a rate of 3–5 °C/min, followed by rapid cooling to −80 °C at 10–40°C/min, and a final descent to −150°C at a rate of ≥50 °C/min. The entire procedure spanned approximately 3–5 min. Immediately following completion, the straws were transferred to a liquid nitrogen tank for long-term storage.

### 2.3. Semen Quality Assessment

#### 2.3.1. Sperm Kinematic Analysis

Sperm kinematic analysis was conducted using a computer-assisted sperm analysis (CASA) system. The following motility parameters were evaluated: straight-line velocity (VSL), average path velocity (VAP), and curvilinear velocity (VCL). Twenty-five consecutive digital images were captured and analyzed per field of view. The proportional representation of each sperm subpopulation relative to the total sperm count was subsequently calculated.

#### 2.3.2. Sperm Plasma Membrane Integrity Assay

First, transfer 20 μL of semen into a pre-warmed centrifuge tube. Then gradually introduce 180 μL of preheated hypotonic solution. Gently vortex the mixture to achieve homogeneity, then incubate at 37 °C for 30 min. Subsequently, aspirate 8 μL of the mixture using a micropipette (Eppendorf, Hamburg, Germany), transfer it onto a glass slide, prepare a smear, and allow it to air-dry naturally. Photomicrographs were taken under a phase-contrast microscope (BX53, Olympus, Tokyo, Japan) using a 400× field of view.

#### 2.3.3. Sperm Mitochondrial Activity

Sperm mitochondrial membrane potential was assessed using a modified version of the previously established methodology [26]. Diluted samples were stained with JC-1 (1 mg dissolved in 1 mL DMSO, Cat. No. HY-15534, MCE Express, Monmouth Junction, NJ, USA) and PI (1 mL PI solution prepared in 10 mL ultrapure water, Cat. No. 25535-16-4, MCE Express, NJ, USA). After 30 min of incubation at 37 °C in the dark, 5 μL of the mixture was aspirated using a micropipette, deposited onto a glass slide, and immediately examined under a fluorescence microscope (BX53, Olympus, Tokyo, Japan) following coverslip placement. At least 200 sperm cells from five distinct microscopic fields were evaluated to determine mitochondrial membrane potential.

#### 2.3.4. Assessment of Sperm DNA Integrity

The methodology for assessing sperm DNA integrity was adapted from the referenced protocol with minor modifications [27]. Transfer 95 μL of diluted semen into a pre-warmed centrifuge tube. Rapidly add 3 μL of Acridine Orange (AO) working solution (1 mg AO dissolved in 10 mL ultrapure water, Cat. No. 65-61-2; MCE, NJ, USA) and 4 μL of Ethidium Bromide (EB) working solution (1 mg EB dissolved in 10 mL ultrapure water, Cat. No. 1239-45-8; MCE, NJ, USA), then mix by gentle vortexing. Protect the mixture from light and incubate at 37 °C for 15 min. To terminate the reaction, introduce 8 μL of Hancock’s solution (formulated with 700 μL 37% methanol, 3.5 mL physiological saline, and 5.8 mL ultrapure water) with thorough pipette mixing. Finally, aspirate 4 μL of the final suspension, deposit onto a clean glass slide, and carefully apply a coverslip. Five different fields of view were observed and captured using a fluorescence microscope (Model BX53, Olympus, Tokyo, Japan), and the counts of spermatozoa with different fluorescence in the head region were recorded.

#### 2.3.5. Assessment of Sperm Acrosome Integrity

The chlortetracycline (CTC) assay was performed with modifications based on the method described [28]. Pre-warm the centrifuge tube, add 20 µL of thawed semen, followed by an equal volume (1:1 ratio) of CTC (Cat. No. 64-72-2, MCE, NJ, USA) staining solution. After incubating at 37 °C for 3 min protected from light, add 8 µL of glutaraldehyde solution to fix the spermatozoa. Aspirate 4 µL of the sperm mixture using a micropipette, deposit it onto a glass slide, and examine under a fluorescence microscope. Randomly select and assess a minimum of 400 spermatozoa. Sperm with intact acrosomes showed uniform fluorescence throughout the head, while those with damaged acrosomes exhibited no fluorescence in the head region.

#### 2.3.6. Biochemical Assays

All antioxidant bioassays were performed according to referenced methodologies with appropriate modifications [7,29]. Thawed frozen semen was diluted with PBS and centrifuged to obtain a pellet. The pellet was then resuspended, and cells were lysed using an ultrasonic cell disruptor (FB50, Thermo Fisher Scientific, MA, USA). The resulting homogenate was centrifuged, and the supernatant was collected for subsequent experiments. The resulting homogenate was centrifuged, and the supernatant was carefully collected for subsequent experiments. Glutathione peroxidase (GSH-Px) activity was determined using a commercial assay kit (Cat. No. A005). Total antioxidant capacity (T-AOC) was measured with an FRAP assay kit (Cat. No. A015-3-1). Malondialdehyde (MDA) content, an indicator of lipid peroxidation, was quantified using a thiobarbituric acid (TBA) assay kit (Cat. No. A003-1). All kits were supplied by Nanjing Jiancheng Bioengineering Institute (Nanjing, China).

### 2.4. Bioinformatics and Proteomic Analysis

TMT Labeling: After enzymatic digestion, an appropriate amount of peptides from each sample was vacuum-dried and reconstituted in 100 mM TEAB to a final volume of 30 μL. Each TMT reagent label was dissolved in 100% anhydrous acetonitrile. Peptides were labeled with TMT tags at a TMT reagent-to-peptide ratio of 5:1 (*w*/*w*) and incubated at room temperature for 1–2 h. The labeling reaction was terminated by adding 5% hydroxylamine to a final concentration of 0.4%.

Protein Identification and Quantification: Raw mass spectrometry data from TMT labeling were analyzed using MaxQuant software (version 2.1.4) for protein identification and quantification. The UniProt database for the corresponding species was used as the reference. False discovery rate (FDR) thresholds were set to 1% for both peptide-spectrum matches and protein identification. Contaminant and decoy database proteins were excluded.

Statistical Analysis: Data analysis was performed using R software (version 4.1.2). Protein raw intensity values were normalized by median scaling. Cluster heatmaps were generated with the R package pheatmap, and significantly differentially expressed proteins (DEPs) were identified using the R package metaX. DEPs were defined as those simultaneously satisfying a **t**-test *p*-value < 0.05 and a fold change > 1.2. Functional enrichment analysis based on Gene Ontology (GO), KEGG Pathway, and Reactome Pathway databases was conducted, with functional terms having a statistical *p*-value < 0.05 considered significantly enriched.

The mass spectrometry proteomics data have been deposited to the ProteomeXchange Consortium via the PRIDE partner repository with the dataset identifier PXD071276 [30].

### 2.5. In Vitro Fertilization

#### 2.5.1. Oocyte Preparation

The matured oocytes were removed from the maturation medium and subjected to cumulus cell removal via enzymatic digestion in a solution containing 0.1% hyaluronidase. Subsequently, mature oocytes were washed three times with modified Tris-buffered medium (mTBM, 1 mM caffeine, 0.1% bovine serum albumin). The washed oocytes (40–50 per group) were then transferred into in vitro fertilization (IVF) culture microdroplets and equilibrated for 30 min in an incubator at 39 °C with 5% CO_2_.

#### 2.5.2. Sperm Processing

For semen processing, frozen-thawed semen was washed three times in phosphate-buffered saline (PBS) containing 0.1% BSA by centrifugation at 1900× *g* for 3 min each. The final sperm pellet was resuspended in IVF medium, appropriately diluted, and mixed with an equal volume of IVF medium containing oocytes at a concentration of 50 µL sperm suspension (approximately 5 × 10^6^ sperm/mL) for co-incubation.

#### 2.5.3. Fertilization and Embryo Culture

Fertilization was conducted under standard culture conditions (39 °C, 5% CO_2_, saturated humidity) for 4–6 h. After fertilization, the oocytes were washed in Porcine Zygote Medium-3 (PZM-3) to remove attached sperm and transferred into pre-equilibrated (2 h) PZM-3 culture microdroplets (500 µL/drop, 40–50 oocytes/drop) for in vitro culture (IVC) of embryos. The cleavage rate was assessed at 48 h post-fertilization as an indicator of early embryonic development.

### 2.6. Statistical Analysis

All experiments were independently performed on at least three separate occasions. Statistical analysis was conducted using SPSS 26.0 software. Statistical evaluations were performed with GraphPad Prism software (version 9.5). Prior to analysis, data normality was assessed using the Shapiro-Wilk test, and homogeneity of variance was examined using Levene’s test. Data meeting these assumptions were analyzed by one-way analysis of variance (one-way ANOVA). Multiple comparisons were carried out using the Least Significant Difference (LSD) test. All data are expressed as “mean ± Standard Error of the Mean (SEM)”, with significance thresholds defined as follows: *p* < 0.05 denoting a significant difference (*), *p* < 0.01 demonstrating a highly significant difference (**).

## 3. Results

### 3.1. Screening of Semen with Differential Low-Temperature Tolerance

In this study, ejaculates were collected from 37 Large White boars at Shanxi Baba Swine Breeding Company to identify semen samples with varying cryotolerance capacity. Fresh semen samples exhibiting initial motility exceeding 80% were subjected to cryopreservation. A comprehensive post-thaw analysis was conducted to evaluate sperm motility characteristics (vitality and motility) for each individual boar. Samples were subsequently categorized into good freezability ejaculates (GFE) or poor freezability ejaculates (PFE) groups based on their post-thaw motility recovery performance (Figure 1A,B).

Through screening, nine boars were selected for each group: the cryotolerant group and the cryosensitive group. In the cryotolerant group, the post-thaw sperm viability averaged 74.81 ± 0.45%, with a mean motility of 66.49 ± 0.37%. In contrast, the cryosensitive group exhibited significantly lower post-thaw sperm viability of 44.72 ± 0.45% and average motility of 39.61 ± 0.34%. Statistical analysis confirmed significant differences (*p* < 0.01, *n* = 9) in both viability and motility between the two groups (Figure 1C,D).

### 3.2. Differential Low-Temperature Tolerance Semen Quality Analysis

To investigate the relationship between differential cryotolerance and sperm motility, we assessed the kinematic parameters of sperm from both GFE and PFE groups using a Computer-Assisted Sperm Analysis (CASA) system. As shown in Figure 2, the GFE group exhibited significantly higher straight-line velocity (VSL) (Figure 2A; 35.57 ± 1.37% vs. 25.33 ± 0.79%, *p* < 0.01), curvilinear velocity (VCL) (Figure 2B; 58.97 ± 1.37% vs. 45.15 ± 1.98%, *p* < 0.01), and average path velocity (VAP) (Figure 2C; 39.81 ± 1.53% vs. 27.13 ± 1.15%, *p* < 0.01) compared to the PFE group. These results indicated that sperm from the GFE group demonstrate superior resistance to cryodamage, thereby better maintaining their motility characteristics during freeze-thaw cycles.

Cryopreservation-induced damage during freeze-thaw cycles can compromise sperm plasma membrane, DNA integrity, acrosome integrity and mitochondrial activity. This study examined whether cryotolerance differences among sperm are associated with variations in these integrity and functional parameters. As shown in Figure 2, the GFE group exhibited significantly higher plasma membrane integrity (Figure 2D; 43.89 ± 1.15% vs. 27.73 ± 1.32%, *p* < 0.01), DNA integrity (Figure 2E; 60.64 ± 1.34% vs. 30.58 ± 0.79%, *p* < 0.01), acrosome integrity (Figure 2F; 55.39 ± 1.23% vs. 24.75 ± 1.07%, *p* < 0.01), and mitochondrial activity (Figure 2G; 58.18 ± 1.25% vs. 27.81 ± 0.89%, *p* < 0.01) compared to the PFE group. These results indicated that GFE sperm possess enhanced resistance to cryodamage, better maintaining structural integrity of both plasma membranes and DNA during freeze-thaw processes; GFE sperm may possess superior capabilities in completing normal fertilization processes and energy metabolism relative to their PFE counterparts.

### 3.3. Differential Low-Temperature Tolerance Semen Antioxidant Capacity Detection

As shown in Figure 3, cryotolerant sperm exhibited markedly higher T-AOC, SOD and CAT activities (Figure 3A–C; 0.47 ± 0.06% vs. 0.26 ± 0.03%, *p* < 0.01, 25.07 ± 0.47% vs. 19.45 ± 0.29%, *p* < 0.01, 0.92 ± 0.12% vs. 0.57 ± 0.06%, *p* < 0.01), significantly higher GSH-Px (Figure 3D; 203.57 ± 0.24% vs. 173.62 ± 0.42%, *p* < 0.01) activity, and markedly lower MDA levels than cryointolerant sperm (Figure 3E; 3.47 ± 0.63% vs. 6.99 ± 0.21%, *p* < 0.01). This demonstrated superior antioxidant capacity and reduced oxidative stress in cryotolerant sperm, indicating that intrinsic antioxidant capability is a key determinant of differential sperm cryotolerance.

### 3.4. Sperm Proteomics Analysis of the Cryogenic Tolerance Group and the Cryogenic Intolerance Group

To in-depth investigate the differential cryotolerance of boar sperm, this study focused on sperm proteins to analyze their relationship with sperm cryotolerance. By comparing the protein expression profiles of cryotolerant and cryosensitive sperm after freeze-thawing, a total of 1926 sperm proteins were identified. Statistical analysis revealed 22 differentially expressed proteins (DEPs) between the two groups, with 12 proteins significantly upregulated and 10 proteins significantly downregulated (Figure 4A).

Further cluster analysis of sperm proteins from the cryotolerant and cryosensitive groups demonstrated distinct expression patterns of these DEPs between the two groups (Figure 4B). Through bioinformatics analysis, 22 characteristic proteins significantly associated with sperm cryotolerance were screened (Table 1). These proteins may serve as potential biomarkers involved in regulating sperm cryotolerance.

### 3.5. Effects of Adding NT5C1B/ADA to Dilution on the Viability, Viability and Semen Quality of Sperm After Thawing in the Low-Temperature Intolerant Group

The combined supplementation of the extender with 1 µg/mL NT5C1B and 1 µg/mL ADA proteins significantly improved the motility and viability of cryointolerant boar spermatozoa (*p* < 0.05). No significant difference was observed between the combination treatment group and the 1 µg/mL NT5C1B-only group (*p* > 0.05), whereas a significant difference was found between the combination treatment group and the 1 µg/mL ADA-only group (Figure 5A,B; *p* < 0.05). Sperm treated with 1 µg/mL NT5C1B exhibited significantly higher average path velocity (VAP) and highly significantly higher curvilinear velocity (VCL) and straight-line velocity (VSL) compared to the control group (Figure 5C; 23.65 ± 0.67% vs. 28.47 ± 1.28%, *p* < 0.05, 34.44 ± 1.31% vs. 40.52 ± 0.87%, *p* < 0.01, 26.05 ± 1.26% vs. 32.84 ± 1.19%, *p* < 0.01). The NT5C1B-treated sperm exhibited significantly improved plasma membrane integrity (Figure 5D; 24.34 ± 1.08% vs. 29.92 ± 1.25%, *p* < 0.01) and DNA integrity (Figure 5E; 28.46 ± 1.28% vs. 34.41 ± 1.49%, *p* < 0.01) compared to controls. Acrosome integrity was significantly enhanced (Figure 5F; 20.03 ± 1.07% vs. 24.52 ± 1.17%, *p* < 0.05) and mitochondrial activity was highly significantly increased (Figure 5G; 22.87 ± 0.96% vs. 28.76 ± 1.17%, *p* < 0.01) in the NT5C1B-treated group versus the control group. These findings collectively indicate that NT5C1B effectively mitigates cryopreservation-induced damage in cryosensitive sperm and enhances their freezing tolerance.

### 3.6. Effects of NT5C1B Supplementation in Diluents on Sperm Fertilizing Capacity and Embryo Cleavage Rate

As shown in Figure 6, cryopreserved sperm from the two groups were subjected to in vitro fertilization (IVF) with mature oocytes following capacitation treatment. After 4–6 h of fertilization, staining was conducted to evaluate sperm penetration rates (Figure 6A,B). The results demonstrated that the NT5C1B-supplemented group exhibited a highly significant increase in sperm penetration rate compared to the control group (54.21 ± 0.88% vs. 44.51 ± 0.86%, *p* < 0.01). However, the penetration rate in the NT5C1B group remained significantly lower than that of the fresh semen group (59.68 ± 0.89%, *p* < 0.05). After a 5-day culture period, early embryonic cleavage rates were assessed (Figure 6C,D). The NT5C1B-supplemented group showed a significantly higher cleavage rate than the control group (56.16 ± 0.89% vs. 46.45 ± 1.03%, *p* < 0.01), though it was still significantly lower than that of the fresh semen group (61.79 ± 1.38%, *p* < 0.05). These findings indicate that cryopreserved semen supplemented with NT5C1B maintains fertilization capability and demonstrates superior performance to the control group, despite not fully reaching the level of fresh semen.

## 4. Discussion

The cryopreservation cycle significantly compromises semen quality parameters. However, studies on semen cryopreservation of various livestock species indicated that fresh semen with similar initial quality exhibited significant differences in post-thaw sperm motility and kinematic parameters. This study screened semen of 37 boars and selected the cryo-tolerant and intolerant individuals to investigate the difference of boar semen cryotolerance. The results showed that the cryopreservation process caused a decrease in sperm kinematic parameters in both groups. However, as compared to the semen in the cryo-tolerant group, the post-thawed sperm in the cryo-intolerant group exhibited significantly lower sperm viability, plasma membrane integrity, DNA integrity, acrosome integrity, and mitochondrial activity. Studies across various species have consistently shown that high cryotolerance in sperm correlates with superior post-thaw motility and overall quality [31,32].

The abundance of polyunsaturated fatty acids (PUFAs) in boar spermatozoal membranes makes them exquisitely sensitive to reactive oxygen species (ROS), resulting in a heightened susceptibility to oxidative stress relative to other species [33]. To explore the differences in antioxidant capacity of sperm with different cryotolerance, this study examined the antioxidant enzyme activities of sperm. It was found that the antioxidant capacity of the cryotolerant group had significantly higher antioxidant enzyme activities than the cryo-intolerant group. Numerous studies have reported that adding exogenous antioxidants to semen extenders can improve the sperm motility and quality of post-thawed semen. Certain non-antioxidant additives can also enhance the sperm quality of cryopreserved semen by improving the antioxidant capacity of sperm [34,35]. These results suggest that sperm cryotolerance is closely related to the antioxidant status of semen, and the improvement of semen antioxidant status can help enhance cryotolerance of sperm.

To elucidate the underlying molecular mechanisms of different cryotolerance of boar sperm, the proteomic methods was used to analyze the difference of protein profile between the sperm with various cryotolerance boar sperm. A total of 22 differentially expressed proteins were identified, including EIF5A2, HSPA12A, FAF1, KRT8, VPS13A, etc. Among these proteins, seven proteins (HSPA12A, PRDX4, ADA, NT5C1B, VPS13A, CST3, and ACRBP) were associated with sperm cold tolerance. ADA is an adenosine deaminase. Studies showed that the abundance of ADA protein is negatively correlated with semen quality in certain species [36]. Elevated ADA abundance has been detected in the serum of infertile men, indicating that ADA is associated with male infertility, and increased ADA level may indirectly reflect reduced sperm quality [37]. In this study, ADA protein was downregulated in cryo-intolerant sperm, which could be related to the differences in sperm processing methods and species. This suggests that ADA is associated with reduced sperm quality in males. Studies have shown that PRDX4 exists in seminal plasma [38]. PRDX4 is one of the critical components of the seminal plasma antioxidant system and it can protect sperm from detrimental effects of ROS. Proteomic results indicated that PRDX4 was downregulated in cryo-intolerant sperm, suggesting that PRDX4 may play a role in antioxidant defense to protect sperm from oxidative damage.

In this study, the higher abundance of NT5C1B were also observed in cryotolerant sperm compared to cryo-intolerant sperm. The NT5C1B is more abundant in boar sperm with high fertility than in the boar sperm with lower fertility [39]. In bull sperm, endogenous 5′-nucleotidase is located in the anterior head region [40]. After adding CPA, the CPA-tolerant sperm (HCS) group exhibited significantly higher NT5C1B levels than the CPA-intolerant sperm (LCS) group [41]. Additionally, a reduction in sperm motility after cryopreservation is associated with NT5C1B abundance [42,43], indicating that NT5C1B may be associated with male fertility by regulating sperm motility. So NT5C1B could serve as a potential indicator for sperm stress tolerance. The HSPA12A protein is consider to have a protective role in cells, and its encoded protein is associated with accelerated apoptosis [44]. In this study, higher abundance of HSPA12A was observed in cryotolerant sperm, suggesting that HSPA12A could exert a cytoprotective effect by inhibiting apoptosis and resulted in the differences in cryotolerance of sperm.

In current semen cryopreservation research, various exogenous substances are added to freezing extenders to improve post-thaw sperm quality. Therefore, we supplemented two downregulated proteins, NT5C1B and ADA, as exogenous additives to the freezing extender to investigate whether their supplementation could enhance the cryotolerance of cryo-intolerant sperm and improve their post-thaw quality. The addition of 1 µg/mL ADA protein to the freezing extender failed to significantly improve sperm motility or quality in the cryo-intolerant group. However, supplementation with 1 µg/mL NT5C1B protein improved kinematic parameters and quality indicators of cryo-intolerant sperm. This indicates that an appropriate concentration of NT5C1B protein can enhance cryotolerance and improve the quality of cryo-intolerant sperm. This protein may also serve as a potential marker for cryo-tolerant sperm.

Studies have shown that NT5C1B, as a 5′-nucleotidase, catalyzes the conversion of nucleotides such as AMP to adenosine; thus, it may function by maintaining energy homeostasis and reducing oxidative stress during cryopreservation. This study proposes that this mechanism may be achieved through the promotion of nucleotide salvage pathways, helping to maintain ATP levels and improve mitochondrial function, thereby reducing cryodamage (Figure 7). Although the addition of NT5C1B to the freezing extender can enhance the cryotolerance of boar sperm, its kinematic parameters remain lower than those of sperm with higher cryotolerance. This suggests that cryotolerance is a collective effect of multiple proteins and some unknown biological molecules, and these differences are highly likely attributed to the inherent genetic factors of individual animals.

Sperm quality, particularly fertilizing ability, cannot be fully reflected by some sperm kinematic parameters and quality indicators [45,46]. Moreover, the conception rates and litter sizes are the two key indicators to evaluate the fertility of cryopreserved semen. They are also the most critical concerns in practical production [47]. Therefore, this study employed in vitro fertilization and sperm-egg binding assays to evaluate whether the freeze-thaw process or these exogenous additives would affect sperm fertilizing capacity. The sperm penetration rate during sperm-egg binding and the cleavage rate of early embryos post-fertilization were measured. The higher cleavage rate in the NT5C1B group demonstrated that NT5C1B could maintain superior sperm motility and quality during cryopreservation while enhancing fertilizing ability, without adversely affecting early embryonic development. In subsequent studies, we will investigate whether NT5C1B influences post-thaw semen quality in other species and its effects on farrowing rates following in vivo artificial insemination, thereby providing more definitive theoretical and technical references for scientific research and practical applications.

## 5. Conclusions

In conclusion, this study further confirms that significant differences in cold tolerance indeed exist among semen samples from different individuals. NT5C1B not only serves as a reliable biomarker for screening boars with superior freezing tolerance, but more importantly, when used as a cryoprotective additive, it effectively enhances semen freezing outcomes. The significant improvements in post-thaw sperm quality and fertilization capacity observed in cryosensitive sperm highlight the direct application potential of NT5C1B for optimizing semen freezing protocols. This advancement contributes to promoting more reliable and efficient application of frozen boar semen in swine breeding programs, while also facilitating deeper exploration of the intrinsic mechanisms underlying cryodamage in porcine sperm.

## Figures and Tables

**Figure 1 animals-15-03530-f001:**
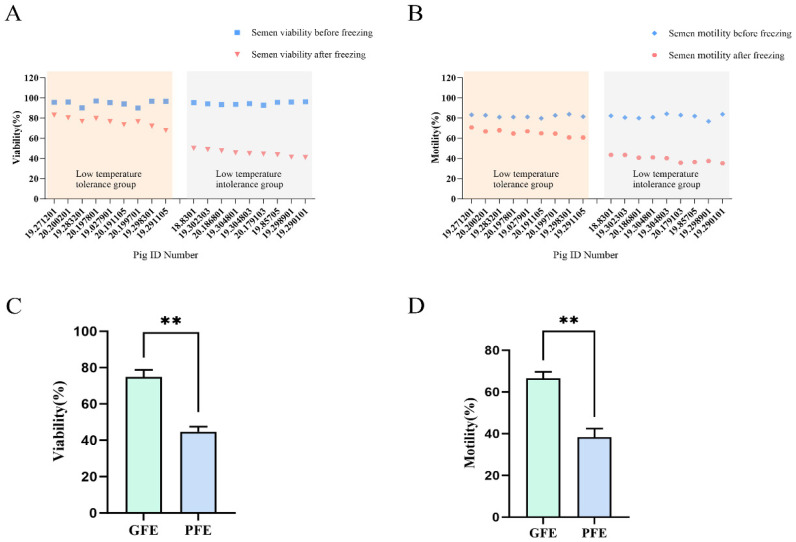
(**A**) Sperm vitality and (**B**) motility of 18 pigs before and after freezing; Sperm (**C**) viability and (**D**) motility of boar frozen-thawed sperm in GFE and PFE. Superscript symbols show statistically differences (** *p* < 0.01).

**Figure 2 animals-15-03530-f002:**
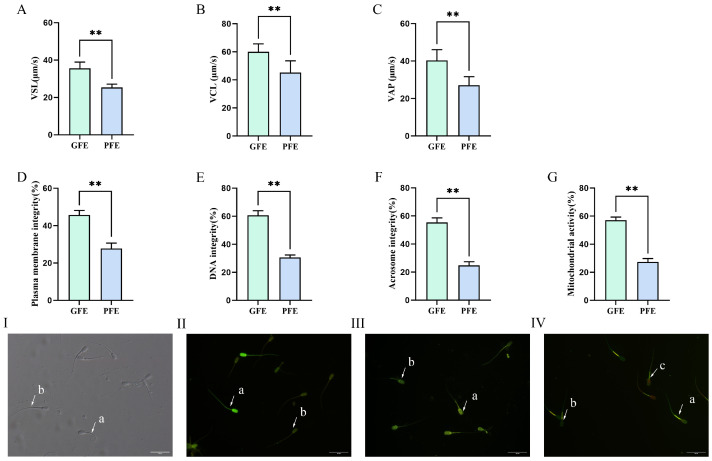
(**A**–**F**) Detection of sperm motility and quality parameters. (**A**) VSL, straight-line velocity; (**B**) VCL, curvilinear velocity; (**C**) VAP, average path velocity; (**D**) plasma membrane integrity, (**E**) DNA integrity, (**F**) acrosome integrity, and (**G**) mitochondrial activity. (**I**): Plasma membrane integrity assay, (a) sperm with intact plasma membrane, (b) sperm with damaged plasma membrane; (**II**): DNA integrity assay, (a) sperm with intact DNA, (b) sperm with fragmented DNA; (**III**): Acrosome integrity assay, (a) sperm with intact acrosome, (b) sperm with damaged acrosome; (**IV**): Mitochondrial activity assay (a) motile sperm with high mitochondrial activity, (b) non-motile sperm with high mitochondrial activity, (c) motile sperm with low mitochondrial activity. ** *p* < 0.01.

**Figure 3 animals-15-03530-f003:**
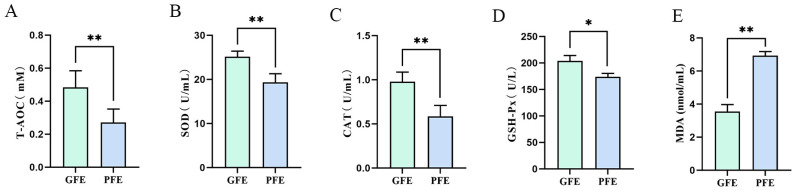
(**A**–**E**) Antioxidant indicators in post-thaw sperm from cryotolerant and cryointolerant groups. (**A**) Total antioxidant capacity, (**B**) superoxide dismutase, (**C**) Catalase, (**D**) Glutathione peroxidase, (**E**) Malondialdehyde. * *p* < 0.05, ** *p* < 0.01.

**Figure 4 animals-15-03530-f004:**
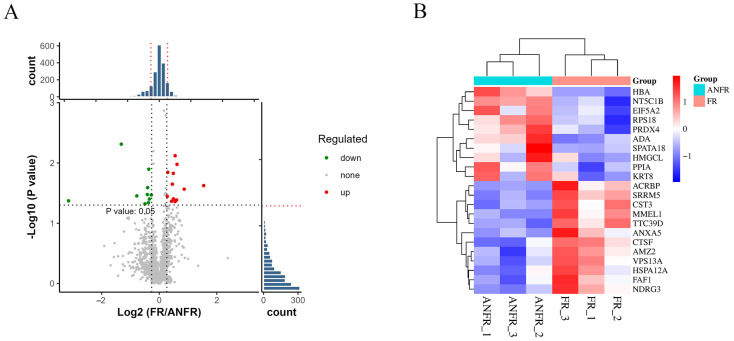
The study of differentially expressed proteins in the proteome related to freeze-tolerance of cryopreserved boar sperm. (**A**) volcano plot of the 1926 quantified DEPs in all biological replicates, (**B**) Differential protein clustering heatmap.

**Figure 5 animals-15-03530-f005:**
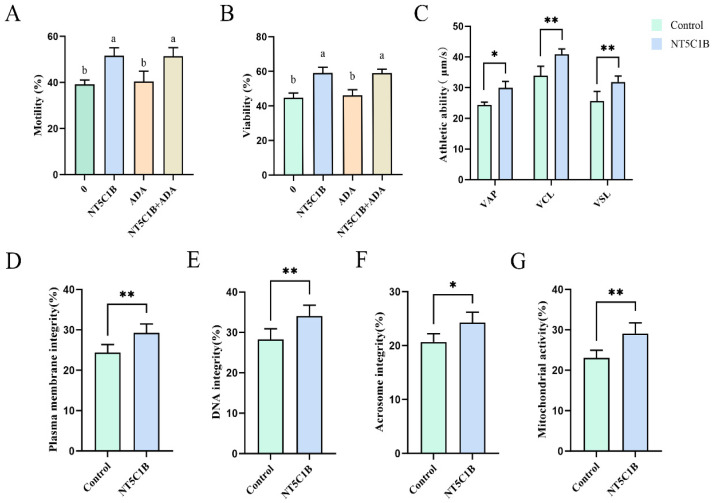
The effects of NT5C1B supplementation on post-thaw cryosensitive sperm were evaluated through the following parameters: (**A**) Combined NT5C1B and ADA supplementation on sperm survival rate, (**B**) Combined NT5C1B and ADA supplementation on sperm motility, (**C**) Combined NT5C1B and ADA supplementation on sperm kinematic parameters, (**D**) Plasma membrane integrity, (**E**) DNA integrity, (**F**) Acrosome integrity, and (**G**) Mitochondrial activity. * *p* < 0.05, ** *p*< 0.01. a, b indicate significant difference groups (*p* < 0.05).

**Figure 6 animals-15-03530-f006:**
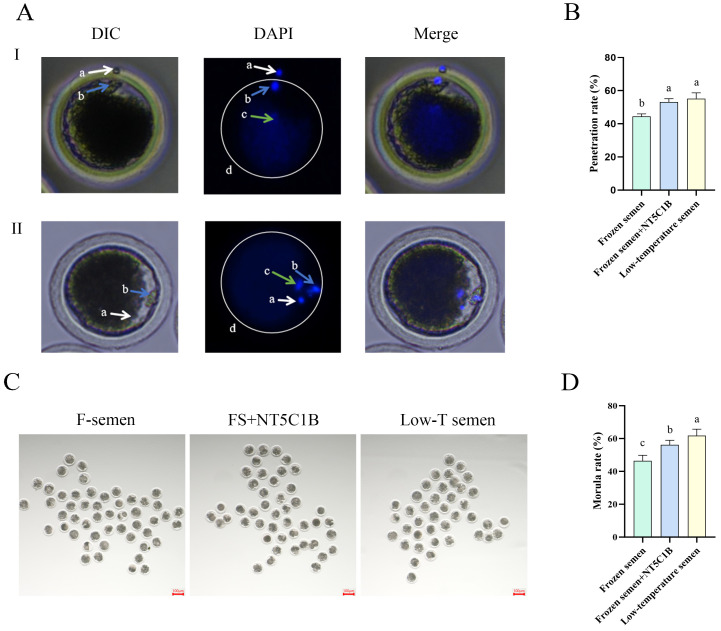
Effect of addition of NT5C1B to dilution on in vitro fertilization of thawed sperm. (**A**) During in vitro fertilization, sperm penetrate the zona pellucida into the oocyte, (**B**) the ratio of sperm passing through the zona pellucida into the oocyte, (**C**) the early embryo cleavage, (**D**) the cleavage rate of early embryos. a: first polar body; b: sperm; c: oocyte nucleus; d: zona pellucida. I: Sperm are passing through the zona pellucida; II: The sperm has already penetrated the zona pellucida. a, b, c indicate significant difference groups (*p* < 0.05).

**Figure 7 animals-15-03530-f007:**
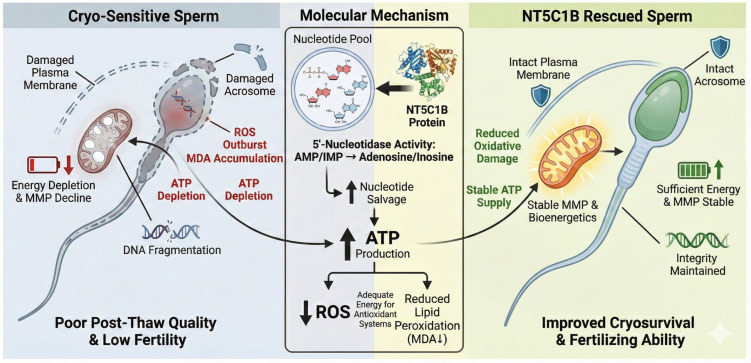
Protective Mechanisms of NT5C1B in Boar Sperm during Cryopreservation. ↑: Enhanced organelle function or increased intracellular substance content; ↓: Decrease in organelle function or reduction in intracellular substance content.

**Table 1 animals-15-03530-t001:** Differentially expressed proteins (DEPs) identified after sperm thawing.

No.	Protein Accessions	Gene Name	Protein Description	Abundance Alteration	log2FC	Biological Function
1	A0A8W4F6B1	SRRM5	Serine/arginine repetitive matrix 5	up	1.544176	Function unknown
2	Q0Z8R0	CST3	Cystatin C	up	0.867238	Signal transduction mechanisms
3	Q29016	ACRBP	Acrosin-binding protein	up	0.618383	Function unknown
4	F2Z5C1	ANXA5	Annexin	up	0.60185	Intracellular trafficking, secretion, and vesicular transport
5	A0A287A1N7	NDRG3	Protein NDRG3	up	0.552163	Function unknown
6	F1S4R7	TTC39D	Tetratricopeptide repeat protein 39B-like	up	0.549316	Function unknown
7	F1RJC1	MMEL1	Membrane metalloendopeptidase like 1	up	0.48997	Amino acid transport and metabolism
8	A0A5G2R698	FAF1	Fas associated factor 1	up	0.489869	Signal transduction mechanisms
9	F1RUT4	CTSF	Cathepsin F	up	0.456895	Posttranslational modification, protein turnover, chaperones
10	A0A287BNS1	HSPA12A	Heat shock protein family A (Hsp70) member 12A	up	0.425675	Posttranslational modification, protein turnover, chaperones
11	F1RV19	AMZ2	Archaelysin family metallopeptidase 2	up	0.297376	Function unknown
12	A0A8W4FJ81	VPS13A	Vacuolar protein sorting 13 homolog A	up	0.279334	Intracellular trafficking, secretion, and vesicular transport
13	A0A8W4FGD5	HMGCL	3-hydroxy-3-methylglutaryl-CoA lyase	down	−0.27517	Energy production and conversion
14	P62272	RPS18	40S ribosomal protein S18	down	−0.3423	Translation, ribosomal structure and biogenesis
15	A0A5G2QBQ4	PRDX4	Peroxiredoxin 4	down	−0.36389	Posttranslational modification, protein turnover, chaperones
16	A0A286ZPS5	EIF5A2	Eukaryotic translation initiation factor 5A	down	−0.37758	Translation, ribosomal structure and biogenesis
17	P62936	PPIA	Peptidyl-prolyl cis-trans isomerase A	down	−0.40374	Posttranslational modification, protein turnover, chaperones
18	A0A8W4F8V4	NT5C1B	5′-nucleotidase, cytosolic IB	down	−0.41346	Function unknown
19	A0A8W4FHJ7	KRT8	Keratin 8	down	−0.49721	Function unknown
20	A0A5G2R146	ADA	Adenosine deaminase	down	−0.77976	Nucleotide transport and metabolism
21	P01965	HBA	Hemoglobin subunit alpha	down	−1.31975	Energy production and conversion
22	F1SE68	SPATA18	Mitochondria-eating protein	down	−3.15223	Function unknown

## Data Availability

The data presented in this study are available on request from the corresponding author.

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
