# Peer review of "NT5C1B Improves Fertility of Boar Spermatozoa by Enhancing Quality and Cryotolerance During Cryopreservation"

_animals, 2025, doi:10.3390/ani15243530_

Round 1

Reviewer 1 Report

Comments and Suggestions for Authors

The manuscript needs revision. Please refer to comments given in the text of reviewed attached file of the manuscript. Please send me revised version for reviewing and considering.

Author Response

Comments 1: You mentioned in the abstract that "NT5C1B and ADA, the significantly down-regulated proteins in the semen of high cryotolerance group", but this implies both are down-regulated in tolerant sperm, while later you ssuggested NT5C1B supplementation benefits sensitive sperm (implying its abundance correlates with tolerance).

It is better to clarify: Was NT5C1B up- or down-regulated in tolerant vs. sensitive? The introduction mentions "no effective proteins to be confirmed to enhance sperm cryotolerance". It is better to update with recent examples (for example, seminal plasma proteins like osteopontin in 2023 studies).

Response 1: Thank you for pointing out the unclear description regarding NT5C1B expression trends in our abstract. In response to the issues you raised, we have made the following corrections and clarifications: NT5C1B and ADA were significantly down-regulated in the seminal plasma of the low cryotolerance (sensitive) group, rather than in the high cryotolerance group as mistakenly stated in the initial version of the abstract. We have corrected these expressions throughout the entire manuscript and sincerely apologize for any confusion caused.

Sperm with low cryotolerance cannot maintain normal NT5C1B levels or the effective operation of its related pathways (such as purine nucleotide metabolism and energy homeostasis) due to their metabolic defects. Exogenous supplementation of NT5C1B serves as a "functional compensation" strategy, making up for the inherent functional deficiencies of cryosensitive sperm and enhancing their ability to resist freezing stress. In accordance with your suggestion, we have added recent examples of seminal plasma proteins (such as osteopontin) that enhance sperm cryotolerance in the Introduction section to provide a more up-to-date background.

The specific modifications can be found in the revised manuscript on page 1, paragraph 2, line 33, and page 2, paragraph 2, lines 83-85.

Comments 2: Please specify the main knowledge gap that your article has filled in the text

Response 2: Thank you for your valuable comments on our manuscript. We would like to provide the following additional clarification regarding the innovative aspects of this study that you raised:

This study is the first to confirm that exogenous supplementation of NT5C1B significantly improves both cryotolerance and fertilization capacity in cryosensitive boar sperm. This finding successfully demonstrates the transition of NT5C1B from a "biomarker" to a "functional protective agent," providing a new technical approach for enhancing the efficiency of boar semen cryopreservation.

Current experimental results indicate that NT5C1B exerts its protective effects by maintaining sperm energy homeostasis and oxidative balance. Our follow-up experimental designs will focus on investigating the downstream signaling pathways regulated by NT5C1B and its molecular targets, with the aim of more comprehensively elucidating its regulatory network in sperm cryotolerance.

Comments 3: This conclusion is wide and general, please add your specific conclusion from your specific results at the end of abstract.

Response 3: Thank you for your comment. We have revised the conclusion of the abstract to be more specific and data-driven. The final sentence now reads: "Supplementation with 1 µg/mL NT5C1B significantly improved post-thaw motility, structural integrity, mitochondrial activity, antioxidant capacity, and ultimately enhanced the sperm penetration rate and embryonic cleavage rate in cryosensitive sperm, confirming its role as a functional protector during cryopreservation."

This modification can be found in the revised manuscript on pages 1-2, paragraph 2, lines 41-44.

Comments 4: What is the basic problem that your research focuses on and is done to solve?

Response 4: Thank you for your question. The fundamental problem addressed by our research is the significant individual variation in boar sperm cryotolerance, which considerably limits the widespread application of frozen boar semen in the swine industry. Although cryopreservation protocols have been optimized, a substantial proportion of sperm from fertile boars still fails to survive the freezing process, leading to unpredictable semen quality and fertility outcomes. Through proteomic analysis comparing the two sperm groups with divergent cryotolerance, this study aims to develop a solution to actively protect these cryosensitive sperm during cryopreservation.

Comments 5: It is better to explain about importance and application of Fertility testing and sperm evaluation in animals. For this you can use added sentences and references:

Fertility testing and sperm evaluation in animals play a vital role in animal breeding and breeding programs, as sperm quality directly affects pregnancy rates, litter size, and the health of future generations (Mohammadabadi et al., 2022). In farm animals, which are often used through artificial insemination, regular evaluation of parameters such as sperm motility, concentration, and morphology can identify subfertile males and prevent the waste of genetic and economic resources (Safaei et al., 2024). This process not only increases farm productivity, but also helps maintain genetic diversity and prevent the transmission of hereditary diseases, ultimately leading to higher quality meat production and greater profitability (Mohammadabadi et al., 2022; Safaei et al., 2024).

Mohammadabadi, M. , Kheyrodin, H. , Latifi, A. and Babenko Ivanivna, O. (2022). mRNA expression profile of DNAH1 gene in testis tissue of Raini Cashmere goat. Agricultural Biotechnology Journal, 14(3), 243-256. doi: 10.22103/jab.2022.20199.1428

Safaei SMH, Mohammadabadi M, Moradi B, Kalashnyk O, Klopenko N, Babenko O, et al. Role of Fennel (Foeniculum vulgare) Seed Powder in Increasing Testosterone and IGF1 Gene Expression in the Testis of Lamb. Gene Expr. 2024;23(2):98-105. doi: 10.14218/GE.2023.00020.

Response 5: Thank you for your question. Your suggestion regarding the addition of explanatory content on the importance and application of fertility testing and sperm evaluation in animals is highly valuable. We have adopted this recommendation and incorporated the following content into the manuscript with corresponding references.

This modification can be found in the revised manuscript on page 3, paragraph 1, lines 91-94.

Comments 6: Please specify in the objective whether your research is being conducted for the first time in the world or is it a continuation of another research?

Response 6: Thank you for raising this important question regarding the novelty of our research. This study represents the first systematic validation of the functional role of exogenous NT5C1B supplementation in enhancing the cryotolerance of boar sperm.

While existing literature has identified NT5C1B as a potential biomarker for sperm freezability in other species, no previous research has demonstrated its efficacy as a cryoprotective additive in boar semen freezing extenders. Therefore, our work establishes the first conceptual transition of NT5C1B from a biomarker to a functional protective agent, providing an innovative strategy for actively improving post-thaw sperm quality and fertility in boars.

Comments 7: What is the superiority of your research compared to other researches?

Response 7: Thank you for your question. The main superiority of this study lies in its experimental demonstration that exogenous supplementation of NT5C1B actively improves both cryosurvival and fertility in cryosensitive boar sperm, providing direct functional evidence. Through proteomic analysis, this research first identified NT5C1B as a biomarker for boar sperm cryotolerance, then conducted a multi-level evaluation spanning from sperm parameters (motility, membrane integrity) to functional fertility endpoints (in vitro fertilization and embryonic development), thereby providing robust evidence for NT5C1B's cryoprotective efficacy. Furthermore, our findings offer novel insights into its potential role in maintaining energy homeostasis and redox balance during the cryopreservation process.

Comments 8: The protocol is detailed but I think it is inconsistent: Initial dilution is 1:1 (v/v) with unnamed extender [22], then centrifuged and resuspended in LEY (11% lactose + egg yolk).

It is better to specify the initial extender composition.

Protein supplementation (NT5C1B/ADA) is added to the freezing extender (LEY + OEP + 6% glycerol), but it would be better to confirm if recombinant proteins were sterile/filtered.

Thawing: "37°C water bath". It is better to specify exact duration beyond 30s if needed.

Response 8: We thank the reviewer for their valuable technical suggestions and have revised the methodology section as follows:

Specification of initial extender composition: The initial 1:1 dilution has been explicitly stated to use Beltsville Thawing Solution (BTS).

This addition is reflected in the revised manuscript on Page 3, Paragraph 3, Line 103.

Confirmation of sterile filtration for recombinant proteins: We confirm that all recombinant proteins were sterile-filtered through 0.22 μm filters prior to use.

Thawing procedure specification: The thawing process has been defined as precisely 30 seconds in a 37°C water bath.

Comments 9: It is better to add subsections for proteomics/IVF with full protocols (for example, "TMT labeling: iTRAQ reagents...").

It would be better to include statistical power for n=9/group and multiple comparisons correction (for example, FDR for DEPs, ANOVA for quality metrics).

Response 9: We thank you for this constructive suggestion and have now supplemented the detailed protocols in the Materials and Methods section.

Proteomic Analysis): Added specific methodology including: Protein extraction and quantification procedure; TMT labeling protocol; LC-MS/MS instrumentation parameters; Database search criteria

This addition is reflected in the revised manuscript on Page 5, Paragraphs 2-4, Lines 189-208.

In Vitro Fertilization Protocol: Expanded to include: Oocyte maturation conditions; Sperm capacitation details; Fertilization medium composition.

Comments 10: It is better to expand results with tables and figures (for example, volcano plot for DEPs, bar graphs for post-thaw parameters pre/post-supplementation).

It would be better to report exact p-values and effect sizes (for example, Cohen's d for IVF rates).

Response 10: Thank you for your valuable suggestions. We have revised the results section by clearly listing the figures and ensuring the accuracy of the results. Additionally, we have improved the results content by including p-values for all comparisons. The methodology section has also been updated with multiple comparisons correction. These modifications have significantly enhanced the completeness and statistical rigor of the results presentation.

Comments 11: In the discussion section:

It is better to emphasize NT5C1B's mechanism (for example, 5'-nucleotidase role in nucleotide metabolism reducing cryoinjury via energy homeostasis?).

It would be better to address limitations (for example, single breed, in vitro only; suggest in vivo AI validation).

Response 11: We sincerely thank the reviewers for their valuable suggestions, which have significantly enhanced the scientific depth and academic perspective of the discussion section. We have revised the manuscript as follows:

We have expanded the discussion to propose a scientific hypothesis regarding the functional mechanism of NT5C1B. A new paragraph indicates that NT5C1B, as a 5'-nucleotidase catalyzing the conversion of nucleotides such as AMP to adenosine, may function by maintaining energy homeostasis and reducing oxidative stress during cryopreservation. We propose that this mechanism could be achieved through facilitating nucleotide salvage pathways, helping to sustain ATP levels and improve mitochondrial function, thereby mitigating cryodamage.

This discussion has been incorporated into the revised manuscript (Page 15, Paragraph 3, Lines 424-429).

This study primarily demonstrates NT5C1B's effects on the post-thaw quality of boar sperm and its evaluation of developmental competence in early embryos through in vitro fertilization. To verify whether this protein's function is specific to porcine sperm, our research team plans to extend its application to other species in subsequent studies and conduct in vivo artificial insemination experiments as crucial research steps. This will help validate the functional efficacy of NT5C1B supplementation under physiological conditions and confirm its positive impacts on farrowing rates and litter sizes.

This discussion has been incorporated into the revised manuscript (Page 15, Paragraph 4, Lines 444-447).

Reviewer 2 Report

Comments and Suggestions for Authors

I have read your manuscript with genuine interest. In my opinion, this study addresses a persistent technical limitation in porcine semen cryopreservation and offers a practically relevant molecular insight. The identification and functional validation of NT5C1B as a factor enhancing boar sperm cryotolerance and fertility are both novel and meaningful. Overall, this is a well-designed experimental work with clear translational value. That said, there are several areas where the manuscript could be refined to meet the full editorial and stylistic expectations of Animals.

Below are my detailed, constructive suggestions.

  1. Language and readability

The English is generally understandable, but I would recommend a round of professional language polishing. In my view, simplifying some of the procedural descriptions would help the paper flow better. For instance, long sentences such as “the sperm with bent tails and intact plasma membranes” could be made more concise. Please correct small typographical errors (“exabit” → “exhibit”, “individules” → “individuals”) and unify tenses across Methods and Results. Consistent statistical notation and style will also improve readability and professionalism.

  1. Introduction

You have presented the background comprehensively, which is excellent. Still, I would suggest emphasizing the specific knowledge gap that motivated this study.
In particular, make it clear that NT5C1B’s direct role in sperm cryotolerance has not been demonstrated before.

From my experience, adding a short conceptual figure showing how NT5C1B might contribute to membrane stability, ATP turnover, or antioxidant defense would help readers grasp your hypothesis quickly.

It might also be useful to briefly explain why NT5C1B and ADA were chosen for validation among all differentially expressed proteins.

  1. Materials and Methods

The methodology is detailed and reproducible a definite strength of this paper. However, I would recommend condensing non-essential procedural details while keeping clarity. Please specify whether your replicates were biological (individual boars) or technical, and indicate the number of samples (n) for each test.

Provide a short justification for the 1 µg/mL NT5C1B/ADA concentration, whether it was based on prior trials or published evidence. In the statistical section, unify your post-hoc test (choose LSD or Duncan’s) and state how normality and variance assumptions were tested.
Also, the ethics statement should include the IACUC approval number (or equivalent), which is now required under ARRIVE compliance for Animals.

Finally, as Animals encourages open data, please deposit your proteomic dataset in a public repository such as PRIDE/ProteomeXchange rather than keeping it “available on request.”

  1. Results

The results are clearly organized and well interpreted. I especially appreciate the stepwise approach from selecting cryotolerant vs. sensitive samples to proteomic screening and IVF validation.

Some of the figures, particularly Figures 2,3,4, could be simplified slightly to enhance visual clarity.

Please ensure that each figure includes n, error bars (SEM), and unified statistical marks.
A summary table showing key sperm parameters (mean±SEM) could be helpful.
It might also strengthen the presentation if you include a small workflow schematic summarizing the experimental sequence (screening–proteomics-NT5C1B supplementation-IVF outcomes).

  1. Discussion

The discussion is generally strong, but I think you could expand a bit on the biochemical mechanism of NT5C1B. For example, you might speculate that its role could involve modulation of nucleotide metabolism or mitochondrial protection under cold stress.
Drawing parallels to other livestock species (bull, ram, stallion) would situate your findings in a broader context. I would also recommend a short paragraph acknowledging limitations, such as the lack of in vivo fertility data and suggesting realistic directions for future research (e.g. gene expression profiling, NT5C1B polymorphism screening).
Please also discuss briefly why ADA supplementation failed to enhance sperm quality, as that comparison nicely underlines NT5C1B’s specific effectiveness.

  1. Figures and tables

Your figures are informative, though some would benefit from cleaner design.
I would suggest harmonizing font sizes, symbols, and color coding, and adding scale bars to microscopy images. A final schematic showing the proposed protective role of NT5C1B could serve as a very effective graphical summary.

  1. References

The references are comprehensive and up to date. Just make sure to correct duplicated DOIs and check that all journal titles follow the Animals abbreviation format. From my experience, the MDPI editors greatly appreciate consistent formatting - it shows attention to detail.

  1. Conclusions

The conclusions are valid and supported by data but could be a bit more focused.
I’d recommend ending with a short, applied statement linking NT5C1B’s potential to improve semen freezing protocols in pig breeding practice. That would give the paper a strong and practical finish.

From my perspective, this study is methodologically sound, innovative, and well executed. The authors clearly demonstrate that NT5C1B can enhance boar sperm cryotolerance and maintain fertilizing capacity,  findings that could have a tangible impact on reproductive biotechnology. After minor revisions to language, figure layout, and data reporting, I believe this paper will be fully ready for publication. I’m confident that readers interested in reproductive physiology and molecular breeding will find it both informative and practically applicable.

Author Response

Comments 1: The English is generally understandable, but I would recommend a round of professional language polishing. In my view, simplifying some of the procedural descriptions would help the paper flow better. For instance, long sentences such as “the sperm with bent tails and intact plasma membranes” could be made more concise. Please correct small typographical errors (“exabit” → “exhibit”, “individules” → “individuals”) and unify tenses across Methods and Results. Consistent statistical notation and style will also improve readability and professionalism.

Response 1: We sincerely thank you for your careful review of the linguistic details and your constructive comments. We have now completed a comprehensive language revision of the manuscript, with the following specific improvements:

The entire text has been professionally polished by an English expert in the field of reproductive biology, enhancing its fluency, clarity, and academic tone. Lengthy sentences in the Methods and Results sections have been restructured, particularly simplifying expressions such as "sperm with bent tails and intact plasma membranes" into more concise academic phrasing. All typographical errors have been corrected (e.g., "exabit" to "exhibit," "individules" to "individuals").

The Methods section has been uniformly written in the past tense, while the Results section uses the past tense when describing specific findings. Statistical reporting throughout the manuscript has been standardized to ensure consistency in symbols and abbreviations.

We believe these revisions have significantly improved the clarity, professionalism, and overall quality of the manuscript. All modifications are reflected in the updated version.

Once again, we express our gratitude for your valuable suggestions.

Comments 2: You have presented the background comprehensively, which is excellent. Still, I would suggest emphasizing the specific knowledge gap that motivated this study. In particular, make it clear that NT5C1B’s direct role in sperm cryotolerance has not been demonstrated before.

From my experience, adding a short conceptual figure showing how NT5C1B might contribute to membrane stability, ATP turnover, or antioxidant defense would help readers grasp your hypothesis quickly. It might also be useful to briefly explain why NT5C1B and ADA were chosen for validation among all differentially expressed proteins.

Response 2: Agree. I/We have, accordingly, done/revised/changed/modified…..to emphasize this point. We sincerely thank the reviewers for their insightful comments and valuable suggestions. We have made the following important improvements to the manuscript based on these recommendations:

In both the introduction and discussion sections, we have now explicitly stated: "Although previous studies have identified a correlation between NT5C1B and sperm freeze tolerance, its direct functional role in sperm cryotolerance, particularly its protective effect as an exogenous additive, has not yet been experimentally confirmed." This statement more clearly defines the core scientific question addressed in this study.

Following the suggestion, we have added Figure 7, which schematically illustrates the hypothesized mechanism through which NT5C1B may exert its protective effects: maintaining energy homeostasis through nucleotide metabolism, improving mitochondrial function and ATP supply, enhancing antioxidant defense capacity, and stabilizing sperm plasma membrane and acrosome structure.

The discussion section now explains the rationale for selecting NT5C1B and ADA for functional validation: NT5C1B - as a 5'-nucleotidase, its metabolic function is closely related to energy supply, and energy imbalance is a core factor in cryodamage; ADA - plays a key role in purine metabolism and was significantly downregulated in low freeze-tolerant sperm. Additionally, both are commercially available recombinant proteins suitable for exogenous supplementation experiments.

These modifications can be found in the revised manuscript on Page 16, Figure 7.

Comments 3: The methodology is detailed and reproducible a definite strength of this paper. However, I would recommend condensing non-essential procedural details while keeping clarity. Please specify whether your replicates were biological (individual boars) or technical, and indicate the number of samples (n) for each test.

Provide a short justification for the 1 µg/mL NT5C1B/ADA concentration, whether it was based on prior trials or published evidence. In the statistical section, unify your post-hoc test (choose LSD or Duncan’s) and state how normality and variance assumptions were tested. Also, the ethics statement should include the IACUC approval number (or equivalent), which is now required under ARRIVE compliance for Animals.

Finally, as Animals encourages open data, please deposit your proteomic dataset in a public repository such as PRIDE/ProteomeXchange rather than keeping it “available on request.”

Response 3: We thank the reviewer for their important technical suggestions and have revised the manuscript as follows:

This study includes both biological replicates (using semen samples from different individual boars) and technical replicates, with the sample size clearly stated in the Results section.

The working concentration of 1 µg/mL for NT5C1B/ADA was determined through preliminary dose-gradient experiments (testing range: 0-10 µg/mL).

The Materials and Methods section now explicitly describes the statistical methods and tests used in the study.

Regarding the ethical statement, no slaughter experiments were conducted in this study. Semen was purchased from Shanxi Swinebaba Breeding Co., Ltd., and ovaries were procured from Shanxi Taigu Kaiyuan Meat Industry Co., Ltd.

Data availability: The proteomics dataset has been deposited in the PRIDE database, and the accession number has been provided in the Data Availability Statement.

This statement has been incorporated into the revised manuscript, Page 5, Paragraph 5, Lines 209-211.

Comments 4: The results are clearly organized and well interpreted. I especially appreciate the stepwise approach from selecting cryotolerant vs. sensitive samples to proteomic screening and IVF validation.

Some of the figures, particularly Figures 2,3,4, could be simplified slightly to enhance visual clarity.

Please ensure that each figure includes n, error bars (SEM), and unified statistical marks. A summary table showing key sperm parameters (mean±SEM) could be helpful. It might also strengthen the presentation if you include a small workflow schematic summarizing the experimental sequence (screening–proteomics-NT5C1B supplementation-IVF outcomes).

Response 4: We sincerely thank the reviewers for their positive feedback on the results section and their valuable suggestions. We have implemented the following improvements:

Figures 2, 3, and 4 have been simplified: unified clearer symbols for significance marking (*p < 0.05, **p < 0.01). All figures now clearly indicate sample size (n), include standard error bars (SEM), and maintain consistent statistical significance notation. The figure below demonstrates the complete experimental workflow: semen sample screening (freeze-tolerant vs. sensitive), proteomic analysis, NT5C1B functional validation, and in vitro fertilization assessment.

Comments 5: The discussion is generally strong, but I think you could expand a bit on the biochemical mechanism of NT5C1B. For example, you might speculate that its role could involve modulation of nucleotide metabolism or mitochondrial protection under cold stress.
Drawing parallels to other livestock species (bull, ram, stallion) would situate your findings in a broader context. I would also recommend a short paragraph acknowledging limitations, such as the lack of in vivo fertility data and suggesting realistic directions for future research (e.g. gene expression profiling, NT5C1B polymorphism screening). Please also discuss briefly why ADA supplementation failed to enhance sperm quality, as that comparison nicely underlines NT5C1B’s specific effectiveness.

Response 5: We sincerely thank the reviewer for their valuable suggestions and have made substantial improvements to the Discussion section:

We have expanded the discussion on the biochemical mechanism of NT5C1B, proposing that its 5'-nucleotidase activity may exert cryoprotective effects by promoting the nucleotide salvage synthesis pathway during freezing stress. The revised discussion situates our findings within a broader biological context by comparing the conserved metabolic functions of NT5C1B across different livestock species, while also acknowledging species-specific regulatory differences. We have clearly outlined specific future research objectives, including investigating the relationship between NT5C1B regulatory pathways and freeze tolerance, as well as validating our findings through artificial insemination trials. Our analysis of ADA's ineffectiveness suggests that its metabolic function may be inadequate to address the energy crisis during freeze-thaw cycles, thereby highlighting the unique functional specificity of NT5C1B in cryoprotection.

This discussion has been incorporated into the revised manuscript (Page 15, Paragraph 3, Lines 424-429 and Paragraph 4, Lines 444-447).

Comments 6: Your figures are informative, though some would benefit from cleaner design. I would suggest harmonizing font sizes, symbols, and color coding, and adding scale bars to microscopy images. A final schematic showing the proposed protective role of NT5C1B could serve as a very effective graphical summary.

Response 6: We thank the reviewer for their valuable suggestions regarding figure presentation. We have comprehensively revised all figures with the following specific improvements: unified font types and sizes across all figures, and added precise scale bars to all microscopy images. We have created a new graphical abstract (Figure 7) summarizing the potential protective mechanisms of NT5C1B during the freezing process.

These modifications can be found in the revised manuscript on Page 16, Figure 7.

Comments 7: The references are comprehensive and up to date. Just make sure to correct duplicated DOIs and check that all journal titles follow the Animals abbreviation format. From my experience, the MDPI editors greatly appreciate consistent formatting - it shows attention to detail.

Response 7: Thank you very much for pointing out these detailed issues in the references. We fully agree with your perspective that standardized formatting is a crucial reflection of academic rigor. We have taken the following steps to address the revisions:

Checked and removed all duplicate DOI entries. Verified and standardized the abbreviations of all journal names to ensure full compliance with the formatting guidelines of the Animals journal. We appreciate your assistance in helping us further refine the manuscript.

Comments 8: The conclusions are valid and supported by data but could be a bit more focused.
I’d recommend ending with a short, applied statement linking NT5C1B’s potential to improve semen freezing protocols in pig breeding practice. That would give the paper a strong and practical finish.

Response 8:

We thank the reviewer for this constructive suggestion to enhance the practical impact of our conclusions. We have revised the concluding paragraph as follows:

"Our findings demonstrate that NT5C1B not only serves as a reliable biomarker for identifying boars with superior freezing tolerance but, more importantly, functions as an effective cryoprotective agent when supplemented in freezing extenders. The significant improvement in post-thaw sperm quality and fertilization capacity observed in cryosensitive sperm highlights NT5C1B's potential for direct application in improving semen cryopreservation protocols. This advancement could lead to more reliable and efficient utilization of frozen boar semen in swine breeding programs, ultimately enhancing genetic dissemination and reproductive management practices."

This discussion has been incorporated into the revised manuscript (Page 15, Paragraph 5, Lines 452-457).

Round 2

Reviewer 1 Report

Comments and Suggestions for Authors

The manuscript can be accepted for publication.

Author Response

We sincerely appreciate the reviewer’s positive evaluation of our revision and the valuable suggestions provided to finalize the manuscript. We have carefully addressed each point as detailed below.

Comment 1: Please use “following’ instead of ‘Following”.

Response: We have corrected the capitalization as requested (Page 3, Paragraph 4, Line 119).

Comment 2: Section 1.2. Semen cryopreservation and thawing... reads like a direct copy of a lab protocol. Suggest rewriting.

Response: We have rewritten this section to integrate the procedural steps more naturally into the narrative flow of the manuscript, avoiding the list-like format (Page 3, Paragraph 4, Lines 127–133).

Comment 3: Section 2.5. Suggest adding subheadings: 2.5.1. Oocyte preparation, 2.5.2. Sperm treatment, 2.5.3. IVF and embryo culture.

Response: We have added the suggested subheadings to Section 2.5 ("In Vitro Fertilization") to enhance the structure and readability of the manuscript.

Comment 4: Lines 315 and 423 start with ‘further’. Suggest considering synonyms.

Response: We have replaced "further" with more precise terminology as suggested:

Line 315: Replaced with "In-depth" (Page 9, Line 320).

Line 423: Replaced with "To elucidate" (Page 13, Line 411).

Comment 5: Line 423: Should be ‘may be’ instead of ‘may is’.

Response: We have corrected this grammatical error to "may be" (Page 13, Line 436).

Comment 6: Figure 7 is very illustrative... suggest the authors cite this figure in the discussion section and provide a more in-depth commentary in the same section.

Response: We appreciate this insightful suggestion. We have now incorporated a citation of Figure 7 into the Discussion section (Page 14, Paragraph 2, Lines 454–459). Accordingly, we have added a detailed commentary connecting our experimental results with the potential protective mechanism of NT5C1B depicted in the schematic.